

**Colloid-bound and dissolved phosphorus species in topsoil water extracts along a grassland**
**transect from Cambisol to Stagnosol**
Xiaoqian Jiang[1], Roland Bol[1], Barbara J. Cade-Menun[2*], Volker Nischwitz[3], Sabine Willbold[3], Sara L.
Bauke[4], Harry Vereecken[1], Wulf Amelung[1,4], Erwin Klumpp[1]
[1] Institute of Bio- and Geosciences, Agrosphere Institute (IBG-3), Forschungszentrum Jülich GmbH,
Jülich, Germany
[2] Swift Current Research and Development Centre, Agriculture and Agri-Food Canada, Box 1030, 1
Airport Rd. Swift Current, SK, S9H 3X2 Canada
[3] Central Institute for Engineering, Electronics and Analytics, Analytics (ZEA-3), Forschungszentrum
Jülich GmbH, Jülich, Germany
[4] Institute of Crop Science and Resource Conservation, Soil Science and Soil Ecology, Nussallee 13,
University of Bonn, 53115 Bonn, Germany
*Corresponding author
Barbara J. Cade-Menun, Email: Barbara.Cade-Menun@AGR.GC.CA




**Abstract**
Stagnant water conditions may release phosphorus (P) in soil solution that was formerly bound to Fe
oxides. To understand which P species are potentially involved, we obtained water extracts from the
surface soils of a gradient from Cambisol, Stagnic Cambisol to Stagnosol from temperate grassland,
Germany. These were filtered to < 450 nm, and divided into three procedurally-defined fractions:
small-sized colloids (20-450 nm), nano-sized colloids (1-20 nm), and "dissolved P" (< 1 nm), using
asymmetric flow field flow fractionation (AF4), as well as filtration for solution $^{31}$P-NMR
spectroscopy. The total P of soil water extracts increased in the order Cambisol< Stagnic Cambisol<
Stagnosol due to increasing contributions from the dissolved P fraction. Associations of C-Fe/Al-PO$_4^{3-}$
/pyrophosphate were absent in nano-sized (1-20 nm) colloids from the Cambisol but not in the
Stagnosol. The $^{31}$P-NMR results indicated that this was accompanied by elevated portions of organic P
in the order Cambisol > Stagnic Cambisol > Stagnosol. Across all soil types, elevated proportions of
inositol hexakisphosphate species (e.g. *myo*-, *scyllo*-, and D-*chiro*-IHP) were associated with soil
mineral particles (i.e. bulk soil and small-sized soil colloids) whereas other orthophosphate monoesters
and phosphonates were found in the 'dissolved' P fraction. We conclude that stagnic properties affect
P speciation and availability by potentially releasing dissolved inorganic and ester-bound P forms as
well as nano-sized organic matter-Fe/Al-P colloids.

**Keywords**: colloidal phosphorus; dissolved phosphorus; field flow fractionation; $^{31}$P-NMR; grassland;
Cambisol; Stagnosol.

**Abbreviations:** AEP, 2-Aminoethyl phosphonic acid; AF4, asymmetric flow field flow fractionation;
Al, aluminum; Ca, calcium; DNA, deoxyribonucleic acid; EDTA, Ethylenediaminetetraacetic; Fe, iron;
FFF, field flow fractionation; ICP-MS, inductively coupled plasma mass spectrometer; myo-IHP,
myo-inositol hexakisphosphate; N,  nitrogen; NMR, nuclear magnetic resonance; OC, organic carbon;
OCD, organic carbon detector; OM, organic matter; PES, polyethersulfone; Pi, inorganic P species; Po,
organic P species; Si, silicon; UV, ultraviolet; WDCs, water dispersible colloids; WDFCs, water
dispersible fine colloids.





**1. Introduction**
Phosphorus (P) is an essential nutrient element for plant growth and limits terrestrial ecosystem
productivity in many arable and grassland soils (Vance et al., 2003). The availability and transport of
P depend on the speciation and concentration of P in the soil solution, which contains both 'dissolved'
and colloidal P forms (Shand et al., 2000; Hens and Merckx, 2002; Toor and Sims, 2015). Dissolved
orthophosphate is generally the main P species in solution and can be directly taken up by plant roots
(Condron et al., 2005; Pierzynski et al., 2005). However, colloidal P in the size range of 1-1000 nm
(Sinaj et al., 1998) may also contribute significantly to total P content in the soil solution (Haygarth et
al., 1997; Shand et al., 2000; Hens and Merckx, 2001). Recent studies found that fine colloids (< 450
nm fraction) in soil water extracts consisted of nano-sized (< 20 nm) and small-sized ($20 < d < 450$ nm)
particles with different organic matter and elemental composition (Henderson et al., 2012; Jiang et al.,
2015a). Very fine nano-sized P colloids, around 5 nm are even prone to plant uptake (Carpita et al.,
1979). In addition, the presence of fine colloids alters the free ionic P content in the soil solution
through sorption processes (Montalvo et al. 2015). After diffusion-limited uptake depletes the free
ionic P in the soil solution, these fine colloids disperse in the diffusion layer and therewith re-supply
free ionic P species for roots (Montalvo et al., 2015). Because water dispersible colloids (WDCs) can
be easily released from soil in contact with water (Jiang et al., 2012; Rieckh et al., 2015), they have
also been suggested as model compounds for mobile soil colloids (de Jonge et al., 2004; Sequaris et al.,
2013). However, little is known about the chemical composition of P species in the different-sized
WDCs.
Recent studies have started to characterize natural fine colloidal P in freshwater samples and soil water
extracts using asymmetric flow field flow fractionation (AF4) coupled to various detectors (e.g.
ultraviolet [UV] and inductively coupled plasma mass spectrometer [ICP-MS]) for improved size
fractionation of colloids and online analysis of their elemental composition (Henderson et al., 2012;
Regelink et al., 2013; Gottselig et al., 2014; Jiang et al., 2015a). These analyses are increasingly
combined with solution [31]P-nuclear magnetic resonance (NMR) spectroscopy, which offers low
detection limits and can quantify different inorganic and organic P compound groups (Cade-Menun,
2005; Cade-Menun and Liu, 2014) in isolated colloidal materials (e.g. Liu et al., 2014; Jiang et al.,





2015a, b; Missong et al., 2016). However, we are not aware of studies that have applied these methods
systematically to WDCs obtained from different major reference soils. Here, we focus on the
comparison of Cambisols and Stagnosols. In contrast to Cambisols, Stagnosols are soils with perched
water forming redoximorphic features. Due to temporary water saturation and resulting oxygen
limitation, the reduction of iron (Fe$^{III}$) is accompanied by the dissolution of its oxides and hydroxides
(Rennert et al. 2014), and the P associated with these Fe-minerals should correspondingly be
redistributed in soil solution.
The objective of this study was to elucidate how stagnant water conditions alter the potential release of
different P compounds in colloidal and 'dissolved' fractions of soil solution. For this purpose, water-
extractable P was obtained from a transect of Cambisols to Stagnosols in a German temperate
grassland, and characterized using both solution $^{31}$P-NMR and AF4 coupled online with UV and
organic carbon detector (OCD) or ICP-MS analyses.

**2. Materials and methods**
**2.1 Site description**
The grassland test site in Rollesbroich is located in the northern part of the Eifel in North Rhine-
Westphalia, Germany (50° 62´ N, 06° 30´ E). According to the soil map of the geological service of
North Rhine-Westphalia (Fig. S1), the dominant soil types on the test site are Cambisols, Stagnic
Cambisols, and Stagnosols (classification according to IUSS Working Group WRB, 2015). The
elevation along the transect generally decreases from south to north, with the highest elevation of
512.9 m a.s.l. at plot 1 and the lowest point of 505.1 m a.s.l. at plot 3 (Fig. S1, Table 1). The
catchment mean annual precipitation was 103.3 cm for the period from 1981 to 2001, and the highest
runoff occurred during winter seasons due to high precipitation and low evapotranspiration rates, as
well as overland flow due to saturation excess (Gebler et al., 2015). The topsoil samples (2-15 cm) of
plot 1 (S1-1, S1-2, and S1-3, Cambisol), 2 (S2, Stagnic Cambisol), and 3 (S3-1, S3-2, and S3-3,
Stagnosol) were taken as a representative transect across the site in early March, 2015 (Fig. S1).
Surface turf (0-2 cm) was removed as it contained predominantly grass roots and little mineral soil.



Stones and large pieces of plant material were removed by hand. All samples were sieved to < 5 mm
and stored at 5 °C for study.

**2.2 Water dispersible fine colloids (WDFCs) separations and AF4-UV-ICP-MS / AF4-UV-OCD**
**analyses**
The WDCs of Rollesbroich grassland soil samples with three field replicates in S1 and S3 were
fractionated using the soil particle-size fractionation method of Séquaris and Lewandowski (2003), but
with moist soils. In brief, moist soil samples (100 g of dry soil basis) were suspended in ultrapure
water (Mill-Q, pH: 5.5) in a soil: solution mass ratio of 1:2, and shaken for 6 h. Thereafter, 600 mL of
ultrapure water were added and mixed. The WDCs suspensions were collected using a pipette after 12
h sedimentation period. These WDCs suspensions were subsequently centrifuged for 15 min at 10,000
$\times$ $g$ and filtered through 0.45 μm membranes to produce the suspension containing WDFCs sized
below 0.45 μm.
An AF4 system (Postnova, Landsberg, Germany) with a 1 kDa polyethersulfone (PES) membrane and
500 μm spacer was used for size-fractionation of the soil sample WDFCs. It is a separation technique
that provides a continuous separation of colloids; the retention time of the colloids can be converted to
hydrodynamic diameters of the colloids using AF4 theory or calibration with suitable standards
(Dubascoux et al., 2010). The AF4 was coupled online to an ICP-MS system (Agilent 7500, Agilent
Technologies, Japan) for monitoring of the Fe, aluminum (Al), silicon (Si), and P contents of the size-
separated particles (Nischwitz and Goenaga-Infante, 2012) and to OCD and UV detectors for
measuring organic carbon (OC). A 25 μM NaCl solution at pH 5.5, which provided good separation
conditions for the WDFCs, served as the carrier. The injected sample volume was 0.5 mL and the
focusing time was 15 min with 2.5 mL min$^{-1}$ cross flow for the AF4-UV-OCD system while 2 mL
injected volume and 25 min focusing time were used for the AF4-ICP-MS system. Thereafter, the
cross flow was maintained at 2.5 mL min$^{-1}$ for the first 8 min of elution time, then set to decrease
linearly to 0.1 mL min$^{-1}$ within 30 min, and maintained for 60 min. It then declined within 2 min to 0
mL min$^{-1}$, and remained at this rate for 20 min to elute the residual particles.



### 2.3 Particle separations of WDFCs and solution $^{31}$P-NMR spectroscopy analyses

The soil samples were treated as described in section 2.2 to obtain the suspension containing WDFCs < 450 nm. We pooled the WDFCs suspensions of the field replicates in order to receive sufficient samples for solution $^{31}$P-NMR. The nano-sized colloidal particles after AF4 separation were smaller than ~20 nm (approximately 300 kDa; Jiang et al., 2015a, Fig. 1). Therefore, the suspension containing WDFCs < 450 nm of these three samples were separated into three size fractions: 300 kDa-450 nm, 3-300 kDa, and < 3 kDa (nominally 1 nm; Erickson, 2009). The 300 kDa-450 nm particle fractions were separated by passing ~600 mL of the WDFCs suspension through a 300 kDa filter (Sartorius, Germany) by centrifugation. The 3-300 kDa particle fractions were subsequently isolated by passing the < 300 kDa supernatant through a 3 kDa filter (Millipore Amicon Ultra) by centrifugation. Finally, the final supernatant containing the < 3 kDa particles as well as the electrolyte phase was frozen and subsequently lyophilized.

The bulk soil samples (1 g) and the three fractions of soil water extracts were respectively mixed with 10 mL of a solution containing 0.25 M NaOH and 0.05 M Na$_2$EDTA (ethylenediaminetetraacetate) for 4 h, as a variation of the method developed to extract soil samples for $^{31}$P-NMR (Cade-Menun and Preston, 1996; Cade-Menun and Liu, 2014; Liu et al., 2014). Extracts were centrifuged at 10,000 × $g$ for 30 min and the supernatant was frozen and lyophilized. Each NaOH-Na$_2$EDTA-treated lyophilized extract, and the < 3 kDa fraction without NaOH-Na$_2$EDTA treatment, was dissolved in 0.05 mL of deuterium oxide (D$_2$O) and 0.45 mL of a solution containing 1.0 M NaOH and 0.1 M Na$_2$EDTA (Turner et al. 2007). A 10 μL aliquot of NaOD was added to the < 3 kDa fraction without NaOH-Na$_2$EDTA treatment to adjust the pH. The prepared samples were centrifuged at 13,200 × $g$ for 20 min (Centrifuge 5415R, Eppendorf). Solution $^{31}$P-NMR spectra were obtained using a Bruker Avance 600-MHz spectrometer equipped with a prodigy-probe (a broadband CryoProbe which uses nitrogen [N]-cooled RF coils and preamplifiers to deliver a sensitivity enhancement over room temperature probes of a factor of 2 to 3 for X-nuclei from $^{15}$N to $^{31}$P), operating at 242.95 MHz for $^{31}$P. Extracts were measured with a D$_2$O-field lock at room temperature. Chemical shifts were referenced to 85% orthophosphoric acid (0 ppm). The NMR parameters generally used were: 32 K data points, 3.6 s repetition delay, 0.7 s acquisition time, 30° pulse width and 10,000 scans. Compounds were identified





by their chemical shifts after the orthophosphate peak in each spectrum was standardized to 6.0 ppm
during processing (Cade-Menun et al., 2010; Young et al., 2013). Peak areas were calculated by
integration on spectra processed with 7 and 2 Hz line-broadening, using NUTS software (2000 edition;
Acorn NMR, Livermore, CA) and manual calculation. Peaks were identified as reported earlier (Cade-
Menun, 2015), and by spiking a select sample with myo-inositol hexakisphosphate (myo-IHP;
McDowell et al., 2007).

**2.4 Statistical Analyses**
Elemental concentrations in bulk soils, soil water extracts, and AF4 fractograms of soil colloidal
particles were tested for significant differences (set to $P < 0.05$) using Sigmaplot version 12.5. A t-test
was conducted to determine the significance of differences among soil sites, whereas one-way
Repeated Measurements (RM) ANOVAs with Fisher LSD were performed with Fisher LSD post-hoc
test to test foridentify significant differences among soil fractions and AF4 fractograms for the
Cambisol and Stagnosol. Data were previously tested to meet the criteria of normal distribution and
homogeneity of variances; for those which had unequal variances data were $\log_{10}$- transformed before
statistical analyses.

**3. Results and discussion**
**3.1 Colloid and colloidal P distribution in different size fractions based on AF4-fractograms**
The AF4-UV-OCD and AF4-ICP-MS results of the WDFCs showed different OC, Si, P, Fe, and Al
concentrations in different-sized colloid fractions as a function of elution time (Fig. 1). The calcium
(Ca) results were not shown because of the generally low colloidal Ca content in these acidic soils.
Before the first peak, an initial small void peak occurred at 1 min (Fig. 1 D, E, F). Thereafter, three
different colloid-size fractions occurred individually as three peaks in the WDFCs of all samples (Fig.
1). The first peak of the fractograms corresponded to a particle size below 20 nm according to the
calibration result using latex standards (Jiang et al., 2015a). The third peak, which was eluted without
cross flow, contained only small amounts of residual particles or particles possibly previously attached
on the membrane during focus time; it had similar OC and element distributions as the second peak in





all samples (Fig. 1). Therefore we considered these two fractions together as a whole. As such, the size
ranges from 20 to 450 nm from here onward are described as the "second size fraction".
For the first fraction representing nano-sized colloids of the three field sites, the OCD and UV signals
indicated increasing OC concentration in the order of S1 (Cambisol; Fig. 1A), S2 (Stagnic Cambisol;
Fig. 1B), and S3 (Stagnosol; Fig. 1C). Distinct peaks of Fe, Al, and P in the first size fraction (< 20 nm)
were only present in the Stagnosol (S3; Fig. 1 F), suggesting that under stagnant water conditions,
oxides may more readily be involved in nano-sized soil particles than under other soil conditions. In
contrast, negligible amount of P, Al, and Fe were detected in the first fraction of S1 and S2 (Fig. 1 D
and E, Table S1). While it is sometimes difficult to determine whether this peak is real or just the
tailing of the void signal (Fig. 1 D and E), solution $^{31}$P-NMR results confirmed the presence of P in
this size fraction (see next section). The nano-sized colloids from the Cambisol contained OC and
negligible P, Fe, and Al; those from the Stagnosol contained significantly higher concentrations of OC,
P, Fe, and Al (Table S1). We therefore assumed that the nano-sized colloidal P forms in the Stagnosol
mainly consisted of OC-Fe(Al)-P associations. Nanoparticulate humic (organic matter)-Fe (Al)
(hydr)oxide-phosphate associations have recently been identified both in water and soil samples
(Gerke, 2010; Regelink et al., 2013; Jiang et al., 2015a). Our results suggest that the formation of these
nano-sized specific P-associations is favoured by the stagnant water conditions with high OC and
water contents in Stagnols but not in the other soil types along the grassland transect.
The second size fraction (Fig. 1 A, B, C, i.e. the small-sized colloids) contained significantly more OC
than the smaller nano-sized colloids for all studied soils (Table S1). Notably, the OC contents of the
second fraction increased in the order Cambisol < Stagnic Cambisol < Stagnosol; the UV signal
therein supporting the results obtained with the OC detector. The larger-sized colloids were
significantly richer in Al, Fe, Si, and P than the smaller-sized ones (Table S1), though again with
differences among subsites: now the stagnic Cambisol showed the largest Fe, Al, and Si contents in
the second fraction, as if there were a gradual change from low WDFC release in the Cambisol to the
formation of larger WDFC in the stagnic Cambisol and finally to the formation of smaller WDFC in
the Stagnosol. Though this trend warrants verification by more sites, it appeared at least as if the
increasing oxygen limitation from Cambisols via stagnic Cambisols to Stagnosols promoted an





increasing formation of small C-rich P-containing nanoparticles with additional contributions from Fe-
and Al-containing mineral phases. Stagnosols like S3 are characterized by a dynamic reduction regime
with dissolution of reactive Fe oxides (Rennert et al. 2014), which leads to a decrease in the content of
Fe oxides in the second colloidal fraction (Table S1). Correspondingly, the dissolution of Fe oxides in
the second fraction under stagnant water may also liberate OC from the organo-Fe mineral
associations, thus releasing OC to the nano-sized first fraction. This could be an additional reason for
the higher concentration of OC in the first peak of S3 (Table S1), apart from a generally slower
degradation of organic matter under limited oxygen supply (Rennert et al. 2014). Hence, the AF4
results indicated that the composition and distribution of particulate P varied among the different-sized
colloidal particles, and that its properties were impacted by the soil type and related properties.
However, AF4-ICP-MS results do not provide information about the elemental concentrations of the
'dissolved' P fraction of these grassland soils.

**3.2 Soil total, colloidal and dissolved P contents based on fractionation by filtration**
Soil water extracts < 450 nm, < 300 kDa, and < 3 kDa were obtained by filtration for determination of
total elemental contents by ICP-MS analysis. Data did not have to be pooled for these analyses; as
such, we could test statistical differences. We considered the soil water extract < 3 kDa in this paper to
be the 'dissolved' fraction. Significant differences ($p < 0.05$) were ascertained for elevated
concentrations of TOC, total P and Ca, as well for lower concentrations of total Al and Fe in the
Stagnosol relative to the Cambisol (Table 1). Furthermore, the Stagnosol had significantly higher
concentrations of Si and P in the individual size fractions of soil water extracts (except marginally
significantly higher P in < 3 kDa, $p = 0.06$), as well as higher Fe and Al concentrations in < 300 kDa
and < 3kDa fraction than the corresponding fractions of the Cambisol (Table 2). The stagnic Cambisol
generally resembled the Cambisol rather than the Stagnosol in bulk soil analysis, but this was not the
case for the soil water extracts. This implied that the assignment of stagnic properties is related to its
behaviors in the colloidal particles and 'dissolved' fraction.
The oxygen limitation and reduction regime of the Stagnosol probably also favored the accumulation
of OC and dissolution of Fe oxides both in bulk soil and colloids (Rennert et al. 2014). Dissolution of



Fe oxides in turn results in a disaggregation of colloidal particles (Jiang et al., 2015a). As the released
oxides are main carriers for P, these processes may explain why the distribution of colloidal and
dissolved P also changed across the different grassland soils. As Table 2 shows, large proportions of P
in the < 450 nm fraction of the Stagnosol were dissolved P (i.e. recovered here in the < 3 kDa fraction),
whereas colloidal P dominated in the Cambisol and Stagnic Cambisol.

**247   3.3 Inorganic and organic P species in the different-sized soil colloidal and the 'dissolved'**

**248   fractions**

Solution $^{31}$P-NMR was used to elucidate the speciation of P in bulk soil and soil water extracts
separated by ultrafiltration into the size fractions 300 kDa-450 nm, 3-300 kDa, and < 3 kDa for each of
the three soils (Fig. 2 and S2, Table 3). The identified P included inorganic P (Pi) forms
(orthophosphate, pyrophosphate, and polyphosphate), and organic P (Po) in phosphonate,
orthophosphate monoester and diester compound classes. Phosphonates included 2-aminoethyl
phosphonic acid (AEP) and several unidentified peaks (Table S3). Orthophosphate monoesters
included four stereoisomers of inositol hexakisphosphate (*myo*-, *scyllo*-, *neo*-, and D-*chiro*-IHP),
diester degradation products (α-glycerophosphate, β-glycerophosphate and mononucleotides), choline
phosphate, and unidentified peaks at 3.4, 4.2, 4.7, 5.0, 5.3, and 5.9 ppm. Orthophosphate diesters were
divided into deoxyribonucleic acid (DNA) and two categories of unknown diesters (OthDi1 and
OthDi2, respectively). Orthophosphate, pyrophosphate, orthophosphate monoesters, and diesters have
also been detected in other studies of grassland, arable, and forest Cambisols and Stagnosols (e.g.,
Murphy et al., 2009; Turrion et al., 2010; Jarosch et al., 2015).
For the bulk soil samples and colloidal fractions of 300 kDa-450 nm of our soil samples,
orthophosphate and orthophosphate monoesters (mainly *myo*-IHP) were the main P compounds in all
samples (Fig. 2 and S2, Table 3 and S2). These main P compounds in these two soil fractions showed
similar trends among the soil samples: the proportions of Po (e.g. orthophosphate monoesters and
diesters) decreased in the order of Cambisol > Stagnic Cambisol > Stagnosol (Table 3). The similarity
in this trend for the different organic P forms can likely be attributed to similarities in the mineral
components of bulk soil and colloidal fractions: i.e., similar element concentrations and thus likely





also similar clay mineralogy, Fe oxide signature and OC content of bulk soil and respective colloid
fraction according to the AF4-OCD and AF4-ICP-MS results (Fig. 1 and Table S1). Orthophosphate,
orthophosphate monoesters and diesters are predominantly stabilized by association with these mineral
components (Solomon and Lehmann, 2000; Turner et al., 2005; Jiang, et al, 2015a). We assume that
the relatively higher proportion of orthophosphate and lower percentage of Po in the Stagnosol may be
attributed to the dissolution of Fe oxides, which likely released Po for mineralization (Condron et al.,
2005). Additionally, the higher concentrations of OC in both bulk soil (Table1) and large colloids of
the Stagnosol probably favored the formation of OC-Fe/Al-PO$_4^{3-}$ complexes (see above).
Our study is the first to distinguish the chemical P composition in colloidal fractions of 3-300 kDa and
300 kDa-450 nm. We found different P speciation and distribution between these two fractions. This is
probably related to differences in their element composition, which are dominated by OC-P/ OC-
Fe(Al)-P associations in the 3-300 kDa soil fraction and by clay-Fe oxides-OC-P associations in the
300 kDa-450 nm size fraction (Fig. 1). Intriguingly, we did not find any Po but only Pi in the 3-300
kDa of all three soils (orthophosphate in Cambisol and Stagnic Cambisol, orthophosphate and
pyrophosphate in the Stagnosol; Table 3). Furthermore, the Stagnosol nanoparticle fraction 3-300 kDa
had a higher proportion of pyrophosphate than the 300 kDa-450 nm size fraction.
When comparing the solution $^{31}$P-NMR results of the < 3 kDa soil fractions with and without NaOH-
Na$_2$EDTA treatments (Fig. 2 and Fig. S2), we observed that most of the phosphonates, orthophosphate
monoesters and diesters were lost after NaOH-Na$_2$EDTA treatment (Fig. 2 and Fig. S2). There were
two possible explanations: 1) 'dissolved' Po in the NaOH-Na$_2$EDTA solution is sensitive and easily
hydrolyzed to orthophosphate (Cade-Menun and Liu, 2014), or 2) in absence of NaOH-Na$_2$EDTA,
most orthophosphate was removed by adsorption on sedimentary material in the re-dissolved solution
after centrifugation when preparing the samples for NMR analysis (Cade-Menun and Liu, 2014),
resulting in elevated portions of Po in the NMR sample. The second possibility may also explain the
observation that there was no orthophosphate in the 'dissolved' fraction of the Cambisol without
NaOH-Na$_2$EDTA treatment (Fig. S2). Almost all the orthophosphate may have been removed with the
sedimentary phase due to the extremely low concentration of dissolved P in this soil. Therefore, we
will focus on the discussion of results obtained from the < 3 kDa soil fractions without NaOH-





Na$_2$EDTA treatment, as they provide better information on the origin of Po-species than the other
samples that received this treatment.
The composition of P species in the < 3 kDa soil fractions (i.e. "truly" dissolved P) differed among the
three soils (Table 3). The majority of P in the < 3 kDa soil fraction of the Cambisol was Po, comprised
mainly of phosphonates and orthophosphate monoesters. The < 3 kDa soil fraction of the Stagnic
Cambisol contained various P species from all compound classes, including orthophosphate,
orthophosphate monoesters, orthophosphate diesters, pyrophosphate, polyphosphates, and
phosphonates. The < 3 kDa soil fraction of the Stagnosol contained similar P species as the Stagnic
Cambisol, with relatively higher proportions of orthophosphate monoesters and phosphonates, but a
lower proportion of orthophosphate diesters (Table 3). It is worth noting that there were more species
of phosphonates in the < 3 kDa fraction than other fractions of each soil (Fig. 2 and S2). The larger
signal  at ~ 21-23.5 ppm was assigned to AEP (Doolette et al., 2009; Cade-Menun, 2015), which
occurred in both the soil particles and the < 3 kDa fraction. However, the small signals at ~ 36-39 ppm
and 45-46 ppm existed only in the < 3 kDa fraction of soil samples (Fig. 2 and S2). The resonance at
36-39 ppm might be assigned to dimethyl methyl phosphonic acid, based on Cade-Menun (2015).
However, spiking experiments were not conducted to identify peaks in this region, so their specific
identity and origins remain unknown.
The solution [31]P-NMR results showed that P species composition in the two colloidal fractions and the
electrolyte phase differed among all three soil samples, with more phosphonates potentially existing in
the electrolyte phase. However, in the study of Missong et al. (2016), more phosphonates and
orthophosphate diesters were found in colloidal fractions rather than the electrolyte phase of two forest
Cambisols. Missong et al. (2016) used centrifugation while we used filtration to separate these particle
sizes and phases. Additionally, Missong et al. (2016) worked with forest soils while we worked with
grassland soils. McLaren et al. (2015) recently confirmed that the speciation of organic P is markedly
different between high (> 10 kDa) and low (< 10 kDa) molecular weight fractions of soil extracts. In
any case, as both colloidal aggregation and stagnant water conditions paralleled and influenced soil
genesis, it seems reasonable to assume that pedogenesis also affects the redistribution of different P
species among different P colloids and the electrolyte phase.




**3.4 Distribution of orthophosphate monoesters and pyrophosphate**


With variations in overall P species composition, the proportions of certain species of orthophosphate
monoesters were also differently distributed among the investigated fractions of the three soils. For
example, the proportion of various IHP stereoisomers (i.e. *myo*-, *scyllo*-, D-*chiro*-IHP) decreased with
decreasing colloid size (Table S2). This suggests that the majority of IHP was associated with soil
mineral particles but did not exist in the dissolved form in our soil samples. The *myo*-IHP stereoisomer
is the principal input of inositol phosphate to soil in the form of plant material (Turner et al. 2002) and
the other stereoisomers may come from plants or may be synthesized by soil organisms (Caldwell and
Black, 1958; Giles et al., 2015). Inositol phosphate is stabilized mainly through strong adsorption on
the surface of amorphous metal oxides and clay minerals (Celi and Barberis, 2007). Shang et al. (1992)
found *myo*-IHP sorbed onto Al and Fe oxides to a greater extent than glucose 6-phosphate. Several
orthophosphate monoesters such as unknown peaks at 3.4, 4.7 and 5.9 ppm were only detected in the
electrolyte phase of soil samples (Table S2). The differences in orthophosphate monoester species
distribution between soil particles and the electrolyte phase show that soil minerals such as clay
minerals and Fe (Al) oxides are only associated with certain species of orthophosphate monoesters
such as IHP, while other species of orthophosphate monoesters exist only in electrolyte phase. Further
research is warranted to fully understand the factors controlling Po in these different size fractions.
It is worth noting that although the proportion of pyrophosphate in bulk soil was very low, there was
more pyrophosphate in the colloidal and electrolyte phases of the Stagnic Cambisol and the Stagnosol
than in the Cambisol, and mostly in the electrolyte and nano-sized colloidal fraction (Table 3).
Pyrophosphate may be of microbial origin (Condron et al., 2005). Our former study (Jiang et al.,
2015b) indicated that Fe/Al oxides were not the main bonding site for pyrophosphate adsorption in
different-sized fractions of an arable soil. Considering that a high proportion of pyrophosphate (38.5%)
existed in the 3-300 kDa fraction of the Stagnosol, which contained P mainly in OC-Fe(Al)-P
associations (see above), it seems reasonable to assume that pyrophosphate existed as a colloidal OC-
Fe(Al)-pyrophosphate complex. In this regard, the accumulation of pyrophosphate may have been
favored by the larger OC contents in this soil (Fig. 1 C).




This study shows for the first time that P species composition varies among the electrolyte phase and
colloids of different size, with the specific distribution being related to the stagnic water regime of the
soil. It could potentially promote P availability by a mechanism that results in a loss of colloids, thus
providing less surface area for the immediate bonding of inorganic P to minerals, while at the same
time potentially releasing organic P from mineral bonding so that it is more prone to decomposition.
Relating the static differences in P species composition among the different soils and fractions to true
dynamics of P transformations, e.g., by performing controlled mesocosm experiments, now warrants
further attention.

**Appendix A. Supplementary data**
The elemental concentrations in AF4 fractograms, phosphorus spectra and species determined by
solution $^{31}$P-NMR as well as solution $^{31}$P-NMR chemical shifts of the P compounds were shown in
supporting information.

**Acknowledgments**
X. Jiang thanks the China Scholarship Council (CSC) for financial support and acknowledges C.
Walraf and H. Philipp for technical assistance. The authors gratefully acknowledge the support by
TERENO (Terrestrial Environmental Observatories) funded by the Helmholtz Association of German
Research Centers.



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





Table 1 General soil characteristics and concentrations of dissolved organic carbon (TOC), total Fe, Al, P, Ca, and Si in bulk S1 (Cambisol), S2 (Stagnic
Cambisol), and S3 (Stagnosol). The uppercase letters indicate significant differences among soil sites (significant difference of soil site 1 and 3 was tested by t-
test, $p < 0.05$).

| Soil | pH[IV] | Water content (%) | Elevation (m a.s.l.) | TOC (g kg$^{-1}$) | Fe*(g kg$^{-1}$) | Al (g kg$^{-1}$) | P (g kg$^{-1}$) | Ca (g kg$^{-1}$) | Si (g kg$^{-1}$) |
|---|---|---|---|---|---|---|---|---|---|
| S1[I] | 4.90±0.12a | 46.5±2.9 | 512.9 | 35.6±2.3a* | 23.0±1.1a* | 52.6±2.9a | 1.2±0.1a | 1.8±0.1a | 320±7.6 |
| S2[II] | 4.90 | 45.3 | 507.5 | 35.8 | 24.0±0.4 | 54.0±2.0 | 1.3±0.1 | 1.8±0.03 | 320±7.0 |
| S3[III] | 5.36±0.20b | 59.0±7.6 | 505.1 | 71.1±15.1b* | 12.8±0.4b* | 38.7±1.1b | 1.8±0.4b | 3.0±0.5b | 312±12.1 |

[I] The mean of sample S1-1, S1-2, and S1-3 ± standard deviation.
[II] The mean of three replicate sample S2 ± standard deviation.
[III] The mean of sample S3-1, S3-2, and S3-3 ± standard deviation.
[IV] The mass ratio of soil : water = 1:2.5.
* Data were log transformed before t-test analyses because of unequal variances.



Table 2 Concentrations of P, Al, Fe, and Si in soil water extracts < 450 nm, < 300 kDa, and < 3 kDa, respectively. Different lowercase and uppercase indicate significant differences among soil sites and soil fractions, respectively (significant difference of soil sites 1 and 3 was tested by t-test, One Way RM ANOVA for soil fractions with Fisher LSD post-hoc test, P <0.05).

| Soil | TOC (g kg⁻¹) | P (mg kg⁻¹) | | | Al (mg kg⁻¹) | | | Fe (mg kg⁻¹) | | | Si (mg kg⁻¹) | | |
|---|---|---|---|---|---|---|---|---|---|---|---|---|---|
| | < 450 nm | <450nm | <300kDa | <3kDa | <450nm | <300kDa | <3kDa | <450nm | <300kDa | <3kDa | <450nm | <300kDa | <3kDa |
| S1[I] | 0.18 | 0.3±0.1a* | 0.2±0.2a* | 0.1±0.1 | 2.0±0.4A° | 0.6±0.0$^a$aB° | 0.6±0.0$^a$aB° | 2.1±0.5A | 0.2±0.0$^a$aB | 0.2±0.0$^a$a*B | 8.1±0.6aA | 6.8±0.3aB | 6.6±0.4aB |
| S2[II] | 0.17 | 1.3±0.9 | 0.5±0.6 | 0.4±0.3 | 7.3±0.3 | 1.1±0.2 | 1.1±0.2 | 9.2±0.5 | 0.4±0.1 | 0.4±0.1 | 14.1±0.5 | 7.3±0.0$^a$ | 7.8±0.8 |
| S3[III] | 0.23 | 4.4±2.0b* | 3.3±2.7b* | 4.1±2.6 | 4.1±3.1 | 0.7±0.1b | 0.7±0.0b | 4.6±3.3 | 0.4±0.1b | 0.5±0.1b* | 14.6±1.3b | 10.6±2.1b | 11.4±2.5b |

[I] The mean of sample S1-1, S1-2, and S1-3 (Cambisol) ± standard deviation.
[II] The mean of three replicate extracts of sample S2 (Stagnic Cambisol) ± standard deviation.
[III] The mean of sample S3-1, S3-2, and S3-3 (Stagnosol) ± standard deviation.
$^a$ Standard deviation of 0.0 means value <0.05.
* Data were log transformed before t-test analyses because of unequal variances.
° Data were log transformed before One Way RM ANOVA analyses because of unequal variances.





Table 3 the proportion (%) of phosphorus species[a] determined by solution [31]P-NMR for the different
soil fractions of S1 (Cambisol), S2 (stagnic Cambisol), and S3 (Stagnosol).

| Soil fractions | Pi | Po | Ortho-P | Pyro-P | poly | P-mono | P-mono* | P-diest | P-diest* | Phon-P |
|---|---|---|---|---|---|---|---|---|---|---|
| | | | | | | --------%-------- | | | | |
| S1 bulk | 43.4 | 56.6 | 41.2 | 1.5 | 0.7 | 52.9 | 44.5 | 2.2 | 10.6 | 1.5 |
| S2 bulk | 47.8 | 52.2 | 46.4 | 0.9 | 0.5 | 48.6 | 43.7 | 1.4 | 6.3 | 2.2 |
| S3 bulk | 63.7 | 36.3 | 63.0 | 0.2 | 0.5 | 31.2 | 27.0 | 1.5 | 5.7 | 3.6 |
| S1 300 kDa-450 nm | 22.8 | 77.2 | 22.8 | -[y] | - | 56.7 | 49.5 | 11.1 | 18.3 | 9.4 |
| S2 300 kDa-450 nm | 56.8 | 43.2 | 53.1 | 1.0 | 2.7 | 29.9 | 26.9 | 5.2 | 8.2 | 8.1 |
| S3 300 kDa-450 nm | 70.2 | 29.8 | 59.7 | 9.2 | 1.3 | 24.2 | 19.9 | 2.8 | 7.1 | 2.8 |
| S1 3-300 kDa | 100 | - | 100 | - | - | - | - | - | - | - |
| S2 3-300 kDa | 100 | - | 100 | - | - | - | - | - | - | - |
| S3 3-300 kDa | 100 | - | 61.5 | 38.5 | - | - | - | - | - | - |
| S1 < 3 kDa | 13.5 | 86.5 | - | - | 13.5 | 26.9 | 26.9 | 1.9 | 1.9 | 57.7 |
| S2 < 3 kDa | 21.3 | 78.7 | 9.5 | 5.1 | 6.7 | 29.3 | 13.8 | 24.2 | 34.6 | 25.2 |
| S3 < 3 kDa | 22.2 | 77.8 | 8.8 | 6.0 | 7.4 | 29.4 | 27.4 | 8.2 | 10.2 | 40.2 |

[a] inorganic P (Pi), organic P (Po), orthophosphate (Ortho-P), pyrophosphate (Pyro-P), polyphosphate
(poly), orthophosphate monoesters (P-mono), orthophosphate diesters (P-diest), phosphonates (Phon-
P). * recalculation by including diester degradation products (α glycerophosphate, β glycerophosphate,
and mononucleotides) with P-diest rather than P-mono (Liu et al. 2014; Young et al. 2013). [y] below
detection limit.







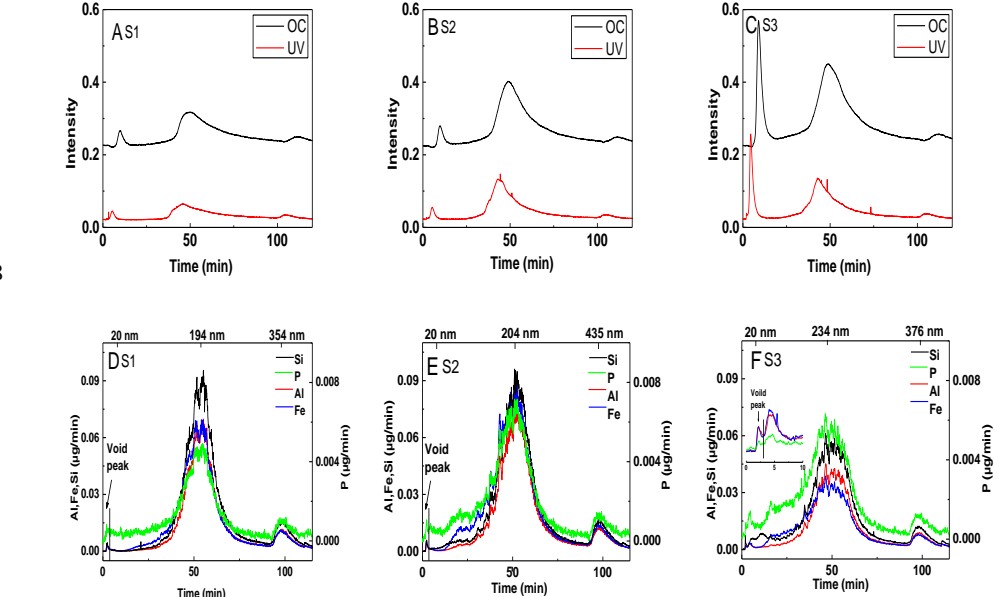

Fig. 1 Asymmetric flow field-flow fractionation (AF4) fractograms of water dispersible fine colloids
(WDFCs) of S1, S2, and S3. The fractograms show the organic carbon (OC) and ultraviolet (UV)
signal intensities (A, B, and C) and the Fe, Al, P, and Si mass flow (D, E, and F) monitored by
inductively coupled plasma mass spectrometer (ICP-MS) of S1 (Cambisol), S2 (Stagnic Cambisol),
and S3 (Stagnosol). The sizes of peaks were according to the AF4 result of sulfate latex standard
particles and dynamic light scattering results. The slight retention time shift between OCD and UV is
due to the internal volume between these two detectors.











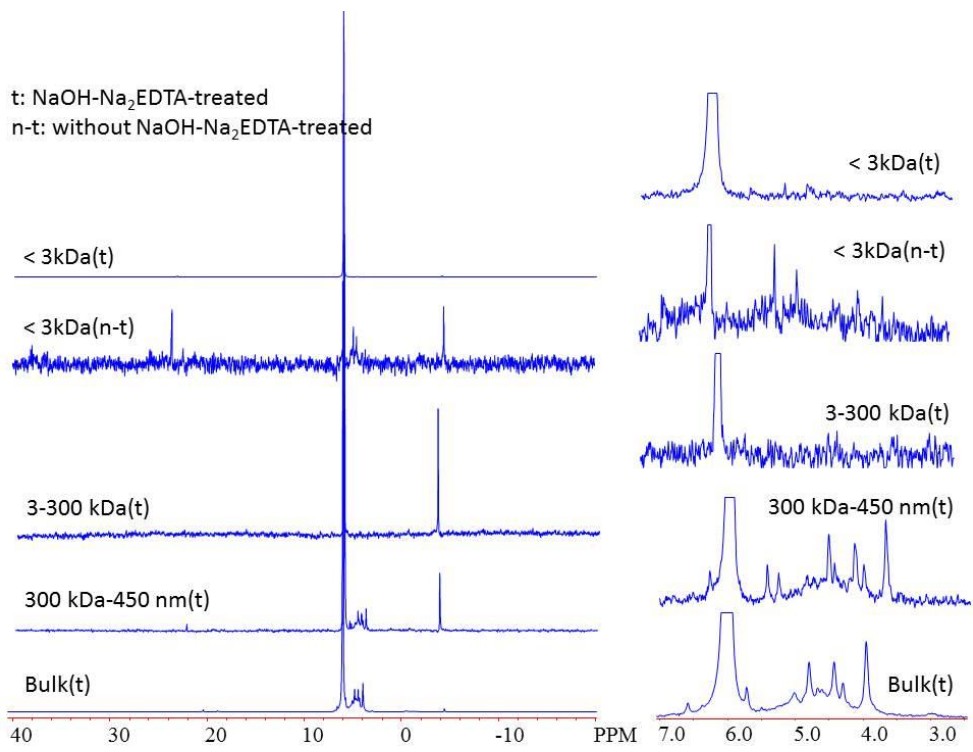


Fig. 2 Solution phosphorus-31 nuclear magnetic resonance spectra of NaOH–Na$_2$EDTA extracts of
bulk soil, 300 kDa-450 nm, 3-300 kDa and < 3 kDa fractions in soil water extracts < 450 nm of S3
(Stagnosol).
