# Peer review of "transect from Cambisol to Stagnosol"

_Biogeosciences, 2016_

## Referee Comment (RC1) · Anonymous Referee #1 · 14 Nov 2016

The manuscript discusses P speciation differences in soil extracts across a landscape gradient with varying soil type, elevation and water content. A number of interesting and quite novel hyphenated techniques were used, which, to the best of my knowledge, have not previously been employed together in such a manner. The ability to differentiate between P species and elemental associations in different colloidal size fractions is technically impressive; particularly as anything less than 0.45 microns in size is very often operationally defined simply as "aqueous P" elsewhere in the literature. The manuscript is easy to follow, well written and logically structured. The mobility and bioavailability of colloidal vs aqueous P is also certainly of broad international interest. However, the research does appear to be quite exploratory and descriptive in

nature rather than hypothesis driven (no hypothesis is stated). The methodology and resulting dataset is very interesting but in my opinion the key conclusion, that soil type and water content affect P speciation in colloidal and aqueous fractions is relatively weak given the variety and complexity of the analytical techniques used. There is also some confusion as to the interpretation and relevance of the results with respect to natural drivers of P mobilisation e.g. iron redox cycling and leaching with pore water transport, due to sample handling which did not preserve field redox conditions. I feel that the manuscript is of interest to a broad audience and suitable for publication in bio-geosciences, however, I think that parts of the manuscript, including the abstract and concluding paragraph, should be carefully and substantially revised to more clearly link the proficient analytical work to the applicability of the results.

Specific comments:

The extract used (MQ water) is quite harsh compared to natural waters such as rain water or pore water and would result in significantly greater release of P than that possible by contact with water in a natural environment, due to desorption and dissolution of poorly crystalline authigenic mineral phases. Living cells within the soil would also certainly undergo significant osmotic stress likely resulting in osmotic rupture and release organic and inorganic P found in intracellular components. The potential ramifications of these effects on the results should be clearly stated and discussed, as there are clearly implications as to the origin and mobility of identified P species in a natural context. Are the species identified in the size fractions indicated present in the natural soil or a result of alterations during the extraction procedure?

I have concerns with the way in which the results are framed within the context of oxygen availability and iron redox cycling. The first sentence of the abstract "Stagnant water conditions may release phosphorus (P) in soil solution that was formerly bound to Fe oxides" implies that the P release investigated is due to reductive dissolution of ferric oxides in the absence of oxygen. Undoubtedly, oxygen availability differences between the soil samples selected resulted in differences to iron speciation, particle size,

organic carbon content and P speciation. The handling of the soil samples in the laboratory does not appear to have preserved the field redox conditions and likely resulted in considerable oxidation of reduced iron species during processing, particularly in the sampled Stagnosols. Oxidation of aqueous $Fe^{2+}$ and colloidal ferrous particles can be very fast (seconds to minutes) therefore the extraction in presumably oxic MQ water for 18 hours almost certainly changed the composition and speciation of the colloids, which were later characterized. Although the importance of Fe oxidation and reduction processes on P speciation generally is highlighted in the manuscript, the impact of these processes during sample processing and on the final dataset is not discussed. The differences between the three soil types are convincing but I question whether the analyzed species and size fractions are representative of the soils themselves or of differences in response to the extraction procedure based on different initial soil redox conditions. Extracting soils in MQ water, under oxic conditions, is not representative of P released during reductive dissolution, as implied in the abstract and in fact would result in the opposite process (oxidative precipitation of Fe hydroxides).

89. Inclusion of a site map would be useful here in the main manuscript rather than in the supporting information. A scale should also be included to establish the distance between the sampling sites.

101. How long were the samples stored at 5oC? Long storage times prior to extraction and preservation could result in significant speciation changes. It is impossible to evaluate the importance of these changes if the storage time is not provided.

112. Please list the material of the 0.45 micron membranes.

119. There is no justification for the choice of analytes - Fe, Al, Si and Ca? The rationale for this may not be clear to some readers.

118. What were the limits of detection and precision for the analytes measured by ICP-MS?

146. Was neutralization of the NaOH-Na2EDTA extracts with HCl performed to avoid break down of polyphosphate species? Perhaps the rationale for not doing so could be included here?

169 – "foridentify" I believe this should read "to test for significant differences" or "to identify significant differences".

170 – Which tests were employed to determine distribution normality?

178 – Analysis of Ca is not previously mentioned. Either calcium analysis should be included in the methods section, and the statement clarified i.e. what constitutes a low concentration? Or this statement could be removed here.

202 – The effect of pH, and differences between the sampled soils should probably be discussed here.

216 – This discussion needs to take into account the effect of the reactions that likely occurred during the oxic extraction procedure.

280 – clay-Fe oxides is an interpretation based on elemental analyses, it is not certain that the colloids identified contain clay minerals from the analyses conducted.

Table 1 – Dissolved or total organic carbon? If this refers to the bulk soil it is not dissolved organic carbon but total organic carbon? The table caption refers to uppercase letters but the letters indicating significant differences are lower case.

Table 2 – To help distinguish between bulk solid analyses and analysis of water extracts I suggest using mg kg-1 for bulk soil analysis and mg L-1 for water analysis. Also – TOC for bulk solid analysis and DOC for aqueous and colloidal analyses.

Table 3 – The formatting and alignment issues make this quite hard to read. "below detection limit" is noted but the detection limit is not defined?

Figure 1 – This figure is great, it really nicely shows the size fractions and their elemental compositions.

[Figure]

---

## Referee Comment (RC2) · Anonymous Referee #2 · 15 Nov 2016

The paper presents interesting and novel data as regards the speciation of P in soil extracts, based on results from three soil samples differing mainly in water saturation conditions. Several innovative techniques are used to characterize the distribution of P species among dissolved and colloidal phases, and most notably the 31P-NMR techniques which provide access to a great number of P organic species. In this respect, the paper is a valuable piece of work as it is the first time that such an inventory of inorganic and organic species is provided for soil extracts, inclijding the colloidal and true dissolved phase. The speciation of P in soil solutions and the way this speciation may vary according to soil properties is clearly of broad international interest, as is of broad international interest the development and application of innovative and accurate analytical techniques to fully characterize the diversity of P forms both in soils and soil solutions. However, I found several weaknesses to the paper that should be addressed and at least partly corrected before its publication in Biogeosciences. The most important are summarized in the following general and more specific comments:

General comments :

Comment 1 : The paper does not give any information about the agricultural use of the sampled grassland sites. Are these sites used for cattle breeding. Do they receive P-containing manure? Are they, or were they, subject to inorganic P fertilization? Were the land use or P fertilization, if any, similar in the cambisol and stagnosol plots? The authors should provide information on those different points.

Comment 2 : Could the authors specify why they use Mill-Q and a pH set at 5.5 in their extraction experiments. Although I can understand that the aim of their study is to compare the behavior of different soils to colloidal extraction and not the impact of the nature of a given extractant and a given pH, I know from my own experience that the nature of the solution and the pH used during colloidal extraction may have great impacts on the composition of the colloids extracted and on the partitioning of P between the colloidal and dissolved phases. Therefore, a justification of the choices made regarding the extracting solution is necessary.

Comment 3 : Too little information is given with respect to the ultrafiltration procedure used for preparing the 31P-NMR spectroscopy samples. First, 600 ml is quite a large volume to ultrafiltrate. Most probably, more than one filter had to be used to ultrafiltrate such big volumes. Could the number of filters used be given? Second, of which material are the filters made of? Are they made of cellulose acetate and if so what are their organic (and phosphorus) blank(s) and the blanks of the overall ultrafiltration procedure?

Comment 4 : Could the authors explain why, in parallel of the ultrafiltration procedure they used to prepare 31P-NMR spectroscopy samples, they also mixed soil water ex-

tracts with 0.025M NaOH and 0.05 M Na2EDTA solution. Was their purpose in doing so to compare these two preparation techniques to check if they were going to give similar (or different) results? Was there a risk that the ultrafiltration technique on its own could fail to give reliable results ?

Comment 5 : The authors should add a size scale in the top diagrams of Fig. 1, in which the OC and UV peaks are portrayed. Anyhow, the first peak on the left of these diagrams seems to occur for a particle size slightly higher than 20. Why then quote in the text (line 181) that the particles corresponding to this first peak are <20 nm in size. Why then also consider in line 186 that only two fractions are higher in particle size than 20 nm? If the same size scale as that shown in the bottom diagrams in Fig. 1 is transposed to the top ones, all the three OC and UV peaks occurring in the latter seem to be for particles of > 20 nm size. Therefore, I am not convinced in the current state of Fig. 1 that the first peak recorded in the analyzed samples correspond to particles below 20 nm.

Comment 6 : In all the top diagrams of Fig. 1 there is a shift between OC and UV peaks, the UV peaks occurring systematically at a lower elution time than the OC ones. What is the reason for this shift?

Comment 7 : I have problems with the idea promoted by the authors that the OC concentration of the first particle fraction would increase from samples 1 to 3. Indeed, the information provided by the top diagrams in Fig. 1 to which they refer lines 188 to 190 are intensities not concentrations. Would it not be equally possible that the OC concentration of the <20 nm particles is in fact constant in the three samples, but that the concentration of this size class of particles in the water extracts increases from sample 1 to 3 ?

Comment 8 : The authors argue that the occurrence of distinct Al and Fe peaks in the first size fraction of the Stagnosol could suggest that oxides are more readily involved in nano-sized soil particles under stagnant soil conditions. I find this interpretation surprising as stagnant conditions are expected to limit the stability of iron oxides. Should it be possible that Fe and Al peaks found in this fraction correspond not to oxides but to Fe and Al ions adsorbed onto, or complexed by organic matter? What proves that the Fe found in this fraction is Fe3+ and not Fe2+. The authors should consider alternative hypotheses of that type here as they do not provide any direct (e.g. spectroscopic) evidence of the presence of Al and Fe oxides in their samples?

Comment 9 : The concentrations quoted in Table S1 are in mg/kg. How these concentrations were calculated ? To what refer Âń kg Âż in this table? To the amount of solution in which the particles are eluted ? To something else ? Please, give precision on that.

Comment 10 : The mechanism promoted by the author of a higher OC concentration in the first peak of the Stagnosol sample due to the release of OC from the larger colloidal fraction because of reductve dissolution of the iron oxides present in this fraction is plausible. However, the AF4 experiments were not performed under reducing conditions. Although this mechanism could probably occur in the true soil solutions under the water saturated conditions that usually prevail in the field in Stagnosol-type soils, I am sceptical about the fact that it could developed in the present case, as the experiments were apparently performed under aerobic conditions.

Comment 11 : The hypothesis brought about by the authors that the oxygen limitation and reduction regime of the Stagnosol would favor the dissolution of Fe oxides in Stagnosol colloids is not entirely convincing to me. Indeed, Table 2 shows higher Fe concentrations in S3 than in S1 sample colloids. Could it be possible that the Fe found in S3 colloids is in part Fe2+ and not Fe oxides?

Comment 12 : I agree with the authors that the dissolution of Fe oxides in the Stagnosol could release Po in the soil solution, but the fate of this Po puzzle me. It seems implicit for the authors that this Po should be readily mineralized and transformed into Pi. However, Stagnosol being waterlogged soils, we expect a reduction of the microbial

activity in these soils and thus of the mineralization rate of Po. I am also not convinced by the hypothesis promoted by the authors on line 276 that the formation of OC-Fe/Al-PO43+ should be favored in these soils. Why should it be so, particularly if iron oxides are expected to be dissolved due to the reducing conditions that characterized Stagnosol as suggested by the authors earlier in the paper. I see a lot of contradictions and approximations here.

Comment 13 : I have difficulties to understand the conclusion of section 3.3 stating that "pedogenesis also affects the redistribution of different P species among different P colloids and the electrolyte phase". I do not see in which respects the results presented in this study allow to put constrains on the pedogenesis of the studied soils and on the impact of this pedogenesis on the presente P speciation results. In my opinion, other variables like land use, anthropogenic P inputs or the methodology used to extract colloids are likely to be as important as, and maybe more important than the pedogenetical history of soils in creating difference in P speciation among soils.

Comment 14: Why pyrophosphate of microbial origin should it be more abundant in Stagnosol than Cambisol, considering that the microbial activity should be enhanced in the more oxygenated Cambisol? I do not pick up authors' arguments here.

Other, more specific comments :

Line 85 : How an organic carbon detector works. Could you specify or quote a refrence in which the principle of the method is described.

Line 116 : Replace " ; " by " . "

Line 119 : Replace " . . . for monitoring of Fe, aluminum (Al). . . " by Âń. . . for monitoring of iron (Fe), aluminum (Al). . . "

Line 132-133. What do the authors mean by "the nano-sized colloidal particles after AF4 separation were smaller than < 20 nm" ? According to Fig. 1, the colloidal particles recovered by the AF4 indeed ranged in size from 20 to 435 nm with peaks at 204 and

435 nm ! Do you mean that the AF4 technique separated all colloids with a nominal size > 20 nm ?

Line 150. I suggest the authors start a new paragraph from "Solution 31P-NMR. . ." as they change of topics from that point, shifting from the description of how the samples were prepared to how the NMR spectra were obtained.

Line 169 : Replace ". . .test to test foridentify. . ." by " . . .test to identify. . . "

Lines 183-184 : What more direct evidence have the authors that the third peaks in the fractograms could correspond to particle previously attached to the mebrane during focus time ?

Lines 195-196 : Is the claim made here that the nano-sized colloids from the cambisol contain P, Fe and Al in lower (negligible) concentrations compared to the same fraction in the stagnosol so true ? Indeed, I calculated the OC/Al and OC/Fe ratios of both soil types and they are not so different : 58 and 93 for Al, and 74 and 105 for Fe. Everything looks like if the nano-sized colloid fractions were equivalent in composition in both soil types, the fraction being simply more concentrated in the stagnosol compared to the cambisol.

Line 202. Replace "Stagnols" by "Stagnosol" Line 205 : I agree that the UV signal is consitent with the OC peak distribution. However, I once again wonder about the reason why the UV peaks are shifted to somewhat lower elution time compared to the OC ones. Could the authors comment on that and provide explanation for this shift ?

Line 208-209 : I agree that the second-size fraction of the stagnic Cambisol present the highest Fe, A, Si and P concentrations of the three analyzed second size-fractions. Considering however ratios of OC/Al, OC/Fe, and OC/P

Line 218 : It is not clear to me why OC sorbed on iron oxides materials should be of nanometric size? Could the authors cite papers which prove this to be so.

Lines 237-238 : "This implies that the assignment of stagnic properties is related to

its behavior in the colloidal particles and dissolved fraction". I find this sentence badly constructed. Do the authors mean that soils are classified according to the composition of the colloids they can release? I cannot believe that.

Line 243 (as regards Table 2): I wonder why TOC concentrations were not measured in the <300 KDa and < 5KDa fractions. Could the authors give an explanation for that? Lines 280-281. I do not see how Fig. 1 can be used to infer the proportion of clay-Fe oxides-OC-P associations in the 300 KDa-450 nm fractions. Could you explain?

Line 300: The statement made here that the majority of P in the <3KDa fraction of the Cambisol was Po is quite "funny" in the light of what is said page 11 about the fact that the absence of NaOH-Na2EDTA most of the Pi is removed from the solution through sorption on the soil mienrals. Quite clearly, the data cannot be used to assess the proportions of Pi and Po as the methodology used biased these proportion. They just can be used to inventory the organic species present, which is already an innovative and very important objective.

Line 348-352. What direct poof do the authors have that pyrophosphates is bound to Fe oxides? Could and alternative interpretation be that orthophosphates form ternary complexes with Fe3+ or Fe2+ ions themselves bound to OC?

---

## Author Comment (AC1) · 30 Jan 2017

**Reviewer 1**

We thank the referee for the detailed comments which helped a lot to improve the manuscript. In this manuscript, we stated a new hypothesis, revised some explanations and conclusion as follows:

Line 18: We changed the sentence "Stagnant water conditions may release phosphorus (P) in soil solution that was formerly bound to Fe oxides" into "Phosphorus (P) species in colloidal and "dissolved" soil fractions may have different distributions".

Lines 31-34: We changed the last sentence into "We conclude that P species composition varies among colloidal and "dissolved" soil fractions after characterization using advanced techniques, i.e. AF4 and NMR. Furthermore, stagnic properties affect P speciation and availability by potentially releasing dissolved inorganic and ester-bound P forms as well as nano-sized organic matter-Fe/Al-P colloids.".

Q: The extract used (MQ water) is quite harsh compared to natural waters such as rain water or pore water and would result in significantly greater release of P than that possible by contact with water in a natural environment, due to desorption and dissolution of poorly crystalline authigenic mineral phases. Living cells within the soil would also certainly undergo significant osmotic stress likely resulting in osmotic rupture and release organic and inorganic P found in intracellular components. The potential ramifications of these effects on the results should be clearly stated and discussed, as there are clearly implications as to the origin and mobility of identified P species in a natural context. Are the species identified in the size fractions indicated present in the natural soil or a result of alterations during the extraction procedure?

A: We agree that a contact of soil to rain and pore water would provide a more realistic scenario; yet, rain and pore water chemistry is variable and thus hard to standardize. As a result, the release of natural nanoparticles from soil could also be variable. Using MQ water for extraction instead of aqueous solutions with higher ionic strength, however, has also two advantages. On the one hand, as also stated by the reviewer, it increases sample dispersion in that we get access to potentially dispersible colloids. We stated this more clearly now. On the other hand, there are analytical advantages, because we avoid interference of additional ions with the retention of particles on the membrane in the channel of FFF, and because MQ water better allows to freeze-dry large amounts of soil solution. Natural water would increase viscosity for the re-dissolved solution, which could increase line broadening and thus decrease the ability to differentiate peak resonances from one another (Cade-Menun and Liu, 2014). We agree that MQ water has potential ramifications of the effects on the mean manuscript as follows:

Lines 123-130: It is worth noting that Mill-Q water was used here to extract soil colloids instead of rain water or pore water, since total amounts of WDFCs will likely be larger when using Mill-Q water, i.e., we consider these WDFCs as potentially water-dispersible colloids. In addition, the use of Mill-Q water facilitates subsequent sample processing with AF4 and NMR. It is inevitable that Mill-Q water would result in the release of P due to desorption

**and dissolution of poorly crystalline authigenic mineral phases. Additionally, living cells within the soil would also certainly undergo significant osmotic stress, likely resulting in osmotic rupture and releasing organic and inorganic P found in intracellular components.**

Q: I have concerns with the way in which the results are framed within the context of oxygen availability and iron redox cycling. The first sentence of the abstract "Stagnant water conditions may release phosphorus (P) in soil solution that was formerly bound to Fe oxides" implies that the P release investigated is due to reductive dissolution of ferric oxides in the absence of oxygen. Undoubtedly, oxygen availability differences between the soil samples selected resulted in differences to iron speciation, particle size, organic carbon content and P speciation. The handling of the soil samples in the laboratory does not appear to have preserved the field redox conditions and likely resulted in considerable oxidation of reduced iron species during processing, particularly in the sampled Stagnosols. Oxidation of aqueous Fe2+ and colloidal ferrous particles can be very fast (seconds to minutes) therefore the extraction in presumably oxic MQ water for 18 hours almost certainly changed the composition and speciation of the colloids, which were later characterized. Although the importance of Fe oxidation and reduction processes on P speciation generally is highlighted in the manuscript, the impact of these processes during sample processing and on the final dataset is not discussed. The differences between the three soil types are convincing but I question whether the analyzed species and size fractions are representative of the soils themselves or of differences in response to the extraction procedure based on different initial soil redox conditions. Extracting soils in MQ water, under oxic conditions, is not representative of P released during reductive dissolution, as implied in the abstract and in fact would result in the opposite process (oxidative precipitation of Fe hydroxides).

A: We agree. However, we also have to annotate hear that stagnant water conditions do not mean that there was stagnant water to the very top of the land surface at time of sampling. By definition, stagnant water dominates for most time of the year and most parts of the soil profile, but it must not (and was) not present in the very surface soil at each time of sampling. When we sampled, the soils were not saturated, i.e., they must have been aerobic already (as common in these surface soils, also in Stagnosols). Hence, the experiment process with Mill-Q water under oxic conditions has potential impact on oxidation of aqueous Fe2+ and colloidal ferrous particles, but we do not see this risk as very severe, because we sampled (and stored) the soils in aerobic conditions. We mentioned it in the manuscript as follows: Line 130-136: It is worth noting that the experimental procedure with Mill-Q water under oxic conditions may have an impact on oxidation of aqueous iron (Fe2+) and colloidal ferrous particles. However, at time of sampling, the very surface soils were not fully water saturated as allowed even for Stagnosols for time of the year. As such, the analyzed species and size fractions are representative of differences in response to the extraction procedure based on different soil redox conditions that reflect a kind of legacy of former redox cycle, but at time of sampling and analyses the soils were aerobic.

Q: 89. Inclusion of a site map would be useful here in the main manuscript rather than in the supporting information. A scale should also be included to establish the distance between the sampling sites.

**A: We added the map with a scale in the main manuscript as follows:**

Fig. 1 Excerpt from the soil map of the test site at Rollesbroich (*modified from Geologischer Dienst Nordrhein-Westfalen, 2008*). Numbered red dots indicate location of plots.

Q: 101. How long were the samples stored at  $5^{\circ}$ C? Long storage times prior to extraction and preservation could result in significant speciation changes. It is impossible to evaluate the importance of these changes if the storage time is not provided.

A: The samples were sieved immediately to < 5 mm and stored at 5°C for less than 6 months before the extraction. All samples were stored in similar manner. The FFF characteristics of WDFCs did not change significantly in the 6 months period of the investigation. We added this information in the manuscript (lines 158-159).

Long storage time under oxic condition have potential impact on the forms of Fe-minerals in soil. However, it is also worth noting that we sampled topsoil (2-15 cm) from Stagnosol which is not the horizon where water is actually stagnating. Additionally, stagnic water conditions do not mean that the soils are under reduced conditions for the whole year – only for some significant time of the year. Although all samples were treated the same way, differences among the samples were consistent with soil characteristics at each site. This suggests that the influences of treatment and storage were minimal.

We gave related discussion as follows:

Lines 103-106: It is worth noting that Stagnic water conditions do not mean that the soils are under reduced conditions for the whole year – only for some significant time of the year. We sampled a Stagnosol, but only the topsoil (2-15 cm) which was not under perching water, i.e., it was aerobic at time of sampling.

Lines 255-259: We cannot rule out any effects from sample storage or from the use of Mill-Q

water, as discussed in the Methods section, However, although all samples were treated the same way, differences among the samples were consistent with soil characteristics at each site. This suggests that the influences of treatment and storage were minimal, but further investigation is warranted in future studies.

Q: 112. Please list the material of the 0.45 micron membranes.

A: The material of membrane was cellulose mixing ester and we added it in the manuscript (line 122).

Q: 119. There is no justification for the choice of analytes - Fe, Al, Si and Ca? The rationale for this may not be clear to some readers.

A: These elements containing minerals (e.g. clay minerals and Fe oxides) were main soil minerals which can be associated with P. We added information about this to the text (lines 145-146).

118. What were the limits of detection and precision for the analytes measured by ICP-MS?

A: The limits of detection (LOD) depend highly on the element, matrix, possible interferences and last but not least the daily performance. The precision, on the other hand, depends mostly on the concentration but also on the element and matrix. Analytes with a concentration close to the LOD have a rather poor precision, whereas higher concentrated analytes achieve a precision of typically 3-10% (relative standard Deviation) depending on the matrix and homogeneity of the samples. The LOD of measured elements in this manuscript is typically around 0.1 to 1 ug/L. We added information about this to the text (lines 157-158).

146. Was neutralization of the NaOH-Na2EDTA extracts with HCl performed to avoid break down of polyphosphate species? Perhaps the rationale for not doing so could be included here?

A: We did not neutralize the NaOH-Na2EDTA extracts with HCl prior to lyophilization, although this was recommended by Cade-Menun et al. (2006, EST 40:7874-7880). Neutralization of samples has not been widely adopted, and was not used by Liu et al. 2014 in their study of WDCs. However, it is something to consider for future studies.

169 – "for identify" I believe this should read "to test for significant differences" or "to identify significant differences".

A: Yes, we changed it into "to test for significant differences among soil fractions" in the manuscript (line 201).

170 - Which tests were employed to determine distribution normality?

**A: We used a Shapiro-Wilks test for normality. This is now indicated in the text (line 202).**

178 – Analysis of Ca is not previously mentioned. Either calcium analysis should be included in the methods section, and the statement clarified i.e. what constitutes a low concentration? Or this statement could be removed here.

**A: We removed this statement.**

202 – The effect of pH, and differences between the sampled soils should probably be discussed here.

**A: Please see our response to Reviewer 2 on this topic.**

216 – This discussion needs to take into account the effect of the reactions that likely occurred during the oxic extraction procedure.

A: We added the following comment on lines 255-259: We cannot rule out any effects from sample storage or from the use of Mill-Q water, as discussed in the Methods section, However, although all samples were treated the same way, differences among the samples were consistent with soil characteristics at each site. This suggests that the influences of treatment and storage were minimal, but further investigation is warranted in future studies.

280 – clay-Fe oxides is an interpretation based on elemental analyses, it is not certain that the colloids identified contain clay minerals from the analyses conducted.

A: We cannot identify clay minerals according to FFF and element analyses. However, clay minerals with Si and Al elements and Fe oxides are common minerals for soils. We did TEM experiments for arable soils in a prior study (Jiang et al., 2015) and found clay minerals and Fe oxides in soil colloids.

Lines 145-146: These elements were analyzed as part of the main soil minerals (e.g. clay minerals and Fe oxides) that can be associated with P (Jiang et al., 2015a).

Table 1 – Dissolved or total organic carbon? If this refers to the bulk soil it is not dissolved organic carbon but total organic carbon? The table caption refers to uppercase letters but the letters indicating significant differences are lower case.

**A: We changed the table caption as suggested.**

Table 2 – To help distinguish between bulk solid analyses and analysis of water extracts I suggest using mg kg-1 for bulk solil analysis and mg  $L^{-1}$  for water analysis. Also –TOC for bulk solid analysis and DOC for aqueous and colloidal analyses.

**A: We changed TOC into DOC as suggested. With respect to the unit of water analysis, we**

**still prefer mg kg-1 because mg L-1 cannot directly tell readers the concentrations of colloidal and dissolved elements compared to those in bulk soil.**

Table 3 – The formatting and alignment issues make this quite hard to read. "below detection limit" is noted but the detection limit is not defined?

A: Below detection limit <0.05%. We added it in the Table 3.

---

## Author Comment (AC2) · 30 Jan 2017

Reviewer 2: We thank the referee for the detailed comments which helped a lot to improve the manuscript. In this manuscript, we stated a new hypothesis, revised some explanations and conclusion as follows:

Q: Comment 1: The paper does not give any information about the agricultural use of the sampled grassland sites. Are these sites used for cattle breeding. Do they receive P-containing manure? Are they, or were they, subject to inorganic P fertilization? Were the land use or P fertilization, if any, similar in the cambisol and stagnosol plots? The authors should provide information on those different points.

[Figure]

A: As we know, the grassland vegetation is dominated by perennial ryegrass (Lolium perenne L.) and smooth meadow grass (Poa pratensis L.). We do not have any information about the amount of P fertilization but we know that there were different managements among these there soils according to personal observation as follows: -for the Cambisols: extensive meadow with three to four cuts per year, no cattle grazing. -for the Stagnic Cambisols: cattle pasture but with less frequent grazing than the Stagnosols -for the Stagnosols: intensively used as pasture with frequent cattle grazing followed by harrowing with a tire-drag harrow and application of organic manure (cattle slurry). We added this information in manuscript (lines 90-96). We also added the following sentence to the discussion of the NMR results (lines 311-316): However, we cannot rule out the effects of differences in grazing and manure application on the P forms in these soils. Cattle grazing and the application of cattle slurry would be expected to add P that is predominantly orthophosphate, with lower concentrations of organic P forms including myo-IHP (Cade-Menun 2011 and references therein). As such, this may have contributed to the increased orthophosphate and decreased organic P we observed on these sites.

Comment 2: Could the authors specify why they use Mill-Q and a pH set at 5.5 in their extraction experiments. Although I can understand that the aim of their study is to compare the behavior of different soils to colloidal extraction and not the impact of the nature of a given extractant and a given pH, I know from my own experience that the nature of the solution and the pH used during colloidal extraction may have great impacts on the composition of the colloids extracted and on the partitioning of P between the colloidal and dissolved phases. Therefore, a justification of the choices made regarding the extracting solution is necessary.

A: Please see our response to Reviewer 1 on this topic.

Comment 3: Too little information is given with respect to the ultrafiltration procedure used for preparing the 31P-NMR spectroscopy samples. First, 600 ml is quite a large volume to ultrafiltrate. Most probably, more than one filter had to be used to ultrafiltrate such big volumes. Could the number of filters used be given? Second, of which material are the filters made of? Are they made of cellulose acetate and if so what are their organic (and phosphorus) blank(s) and the blanks of the overall ultrafiltration procedure?

A: Around 6 filters were used to ultrafiltrate 600 mL solution. The filter was made of regenerated cellulose membrane. Before the ultrafiltration of samples, we washed the filter by filtering Mill-Q water. The P concentration of < 3 kDa fraction of sample 1 is $0.1\pm0.1$ mg/kg, which means there was negligible P concentration in sample 1. It also indicated that the filter material did not affect the P concentration of soil samples. Therefore, although we did not perform the blank experiment, we do not think that there was any P from the filtration. Also according to the NMR results, there was no organic P in 3-300 kDa fractions of soil samples which means there was no potential organic P from the filters into soil samples.

Q: Comment 4 : Could the authors explain why, in parallel of the ultrafiltration procedure they used to prepare 31P-NMR spectroscopy samples, they also mixed soil water extracts with 0.025M NaOH and 0.05 M Na2EDTA solution. Was their purpose in doing so to compare these two preparation techniques to check if they were going to give similar (or different) results? Was there a risk that the ultrafiltration technique on its own could fail to give reliable results?

A: Comment 4: We did not explain it clearly here. The different-sized soil water extracts were obtained by the ultrafiltration procedure and then each size-range soil water extract was mixed to receive sufficient samples for the 31P-NMR characterization. Each size-range soil water extract was then mixed with 0.025 M NaOH and 0.05 M Na2EDTA to extract P for 31P-NMR (see section 2.3). This was done so that all P-NMR experiments were conducted in the sample matrix for bulk soil and soil water extracts.

Q: Comment 5: The authors should add a size scale in the top diagrams of Fig. 1, in which the OC and UV peaks are portrayed. Anyhow, the first peak on the left of these

diagrams seems to occur for a particle size slightly higher than 20. Why then quote in the text (line 181) that the particles corresponding to this first peak are <20 nm in size. Why then also consider in line 186 that only two fractions are higher in particle size than 20 nm? If the same size scale as that shown in the bottom diagrams in Fig. 1 is transposed to the top ones, all the three OC and UV peaks occurring in the latter seem to be for particles of > 20 nm size. Therefore, I am not convinced in the current state of Fig. 1 that the first peak recorded in the analyzed samples correspond to particles below 20 nm. Comment 6: In all the top diagrams of Fig. 1 there is a shift between OC and UV peaks, the UV peaks occurring systematically at a lower elution time than the OC ones. What is the reason for this shift?

A: Comment 5 and 6: Actually the OC and UV peaks occurred with element (ICP-MS) peaks at the same time. The AF4 is connected with UV, OCD and ICP-MS detector with different-length tubes. The slight delay among these peaks is due to the different length of tubes to different detectors which cause slightly different internal volume and retention time (see lines 587-589).

Q: Comment 7: I have problems with the idea promoted by the authors that the OC concentration of the first particle fraction would increase from samples 1 to 3. Indeed, the information provided by the top diagrams in Fig. 1 to which they refer lines 188 to 190 are intensities not concentrations. Would it not be equally possible that the OC concentration of the <20 nm particles is in fact constant in the three samples, but that the concentration of this size class of particles in the water extracts increases from sample 1 to 3 ?

A: Comment 7: We did the calibration of different OC concentrations and found the OC concentrations had a linear positive relation with intensities. It cannot be determined if the concentration of OC or the particles increases from samples 1 to 3. Here the OC concentration is the ratio of OC mass to bulk soil mass but not to <20 nm soil particle fraction.

[Figure]

Comment 8: The authors argue that the occurrence of distinct Al and Fe peaks in the first size fraction of the Stagnosol could suggest that oxides are more readily involved in nano-sized soil particles under stagnant soil conditions. I find this interpretation surprising as stagnant conditions are expected to limit the stability of iron oxides. Should it be possible that Fe and Al peaks found in this fraction correspond not to oxides but to Fe and Al ions adsorbed onto, or complexed by organic matter? What proves that the Fe found in this fraction is Fe3+ and not Fe2+. The authors should consider alternative hypotheses of that type here as they do not provide any direct (e.g. spectroscopic) evidence of the presence of Al and Fe oxides in their samples?

A: Comment 8: It is correct that we need to consider alternative hypotheses with Fe and Al ions besides the iron oxides (see lines 229-231: Nanoparticulate humic (organic matter)-Fe (Al) (ions /(hydr)oxide)-phosphate associations have recently been identified both in water and soil samples (Gerke, 2010; Regelink et al., 2013; Jiang et al., 2015a)). Some published studies have shown the existence of P-Fe/Al-OC complexes with size of ~5 nm (Regelink et al., 2013). In our previous studies with Luvisols (Jiang et al., 2015), we found amorphous Fe/Al oxides in the smaller-sized fractions. However, we did not undertake such specific experiments in the case of Cambisol and Stagnosol.

Q: Comment 9 : The concentrations quoted in Table S1 are in mg/kg. How these concentrations were calculated? To what refer Â'n kg ÂËŹz in this table? To the amount of solution in which the particles are eluted ? To something else ? Please, give precision on that.

A: Comment 9: mg/kg soil particles.

Q: Comment 10 : The mechanism promoted by the author of a higher OC concentration in the first peak of the Stagnosol sample due to the release of OC from the larger colloidal fraction because of reductive dissolution of the iron oxides present in this fraction is plausible. However, the AF4 experiments were not performed under reducing

conditions. Although this mechanism could probably occur in the true soil solutions under the water saturated conditions that usually prevail in the field in Stagnosol-type soils, I am sceptical about the fact that it could developed in the present case, as the experiments were apparently performed under aerobic conditions.

A: Comment 10: It is correct that the experiments were performed under aerobic conditions. However, the AF4 experiment showed the current properties of the Stagnosol and Cambisols. The Stagnosol soils had higher OC concentration in the first peak than the Cambisol. We only say the high OC content in the first peak is apart from the reductive dissolution of iron oxides. A generally slower degradation of organic matter under limited oxygen supply in Stagnosol was another factor for the high OC content. Although the experiments were performed under aerobic conditons, we do not think all the iron ions will be oxidized into iron oxides in our experimental conditions. Stagnic water conditions do not mean that the soils are under reduced conditions for the whole year – only for some significant time of the year. We sampled a Stagnosol, but only the topsoil (2-15 cm) which is not the horizon where water is actually stagnant. As such, the Stagnols used for this study were oxic at various times each year, but also experienced periods of reducing conditions that did not occur in the other samples along the transect. (see lines 103-106).

Q: Comment 11 : The hypothesis brought about by the authors that the oxygen limitation and reduction regime of the Stagnosol would favor the dissolution of Fe oxides in Stagnosol colloids is not entirely convincing to me. Indeed, Table 2 shows higher Fe concentrations in S3 than in S1 sample colloids. Could it be possible that the Fe found in S3 colloids is in part Fe2+ and not Fe oxides?

A: Comment 11: Table 2 shows higher Fe concentrations in S3 but the standard deviation is also extremely high (4.6±3.3), suggesting there were no significant difference between S3 and S1 (2.1±0.5). As shown in the FFF result in Table S1, Fe concentrations in the second peaks in Cambisol and Stagnosol were 7.60±2.11 and 7.34±0.53 mg/kg. It is also possible that the Fe found in S3 colloids were in part Fe2+ absorbed

onto the surface of particles.

Q: Comment 12: I agree with the authors that the dissolution of Fe oxides in the Stagnosol could release Po in the soil solution, but the fate of this Po puzzle me. It seems implicit for the authors that this Po should be readily mineralized and transformed into Pi. However, Stagnosol being waterlogged soils, we expect a reduction of the microbial activity in these soils and thus of the mineralization rate of Po. I am also not convinced by the hypothesis promoted by the authors on line 276 that the formation of OC-Fe/Al-PO43- should be favored in these soils. Why should it be so, particularly if iron oxides are expected to be dissolved due to the reducing conditions that characterized Stagnosol as suggested by the authors earlier in the paper. I see a lot of contradictions and approximations here.

A: Comment 12: The comment about the Po is reasonable and we have no proof that this Po will be readily mineralized and deleted it in the manuscript. The reason why the formation of OC-Fe/Al-PO43- should be favored in Stagnosol soil is that more OC in Stagnosol will bind to more PO43- to form the complex of OC-Fe/Al-PO43-. Although iron oxides are expected to be dissolved in Stagnosol, some Fe oxides are expected to exist in this soil. Stagnic water conditions do not mean that the soils are under reduced conditions for the whole year – only for some significant time of the year. We sampled a Stagnosol, but only the topsoil (2-15 cm) which is not the horizon where water is actually stagnant. On the other hand, Fe/Al ions could also form the complex of OC-Fe/Al-PO43-. (see lines 229-231 )

Q: Comment 13: I have difficulties to understand the conclusion of section 3.3 stating that "pedogenesis also affects the redistribution of different P species among different P colloids and the electrolyte phase". I do not see in which respects the results presented in this study allow to put constrains on the pedogenesis of the studied soils and on the impact of this pedogenesis on the present P speciation results. In my opinion, other variables like land use, anthropogenic P inputs or the methodology used to extract colloids are likely to be as important as, and maybe more important than the

pedogenetical history of soils in creating difference in P speciation among soils.

A: Comment 13: It is correct and we revised it as suggestedïijŽ Lines 361-364: In any case, both colloidal aggregation and changes in soil order paralleled soil P forms. However, also other soil properties but former redox state (like pH), as well as variations in anthropogenic, site-adapted management may be additional covariates affecting P colloids and composition.

Q: Comment 14: Why pyrophosphate of microbial origin should it be more abundant in Stagnosol than Cambisol, considering that the microbial activity should be enhanced in the more oxygenated Cambisol? I do not pick up authors' arguments here.

A: Comment 14: we deleted the sentence: Pyrophosphate may be of microbial origin (Condron et al., 2005).

Q: Line 85 : How an organic carbon detector works. Could you specify or quote a refrence in which the principle of the method is described.

A: The OCD is a promising technique for monitoring organic carbon concentrations for liquid-flow based separation systems with the advantages of high selectivity and low detection limits (Nischwitz et al., 2016). Briefly, the operation principle is that the acidification of the sample flow removes inorganic carbon and subsequently the organic carbon is oxidized in a thin film reactor to carbon dioxide which can be quantified by infrared detection (Nischwitz et al., 2016). We added these sentences in the materials and method section (lines 146-151).

Q: Line 116 : Replace " ; " by " . " Line 119 : Replace " . . .for monitoring of Fe, aluminum (Al). . ." by Â'n: : : for monitoring of iron (Fe), aluminum (Al): : : "

A: We changed the first sentence. However, iron was defined as Fe earlier in the manuscript, so we did not change the second sentence.

Line 132-133. What do the authors mean by "the nano-sized colloidal particles after AF4 separation were smaller than < 20 nm" ? According to Fig. 1, the colloidal particles

recovered by the AF4 indeed ranged in size from 20 to 435 nm with peaks at 204 and 435 nm ! Do you mean that the AF4 technique separated all colloids with a nominal size > 20 nm

A: We defined the first peak fraction as nano-sized colloidal particles. Definitely it misleads readers. We changed the sentence as follows: Lines 164-165: The first peak fraction after AF4 separation has the particle size smaller than ∼20 nm.

Q: Line 150. I suggest the authors start a new paragraph from "Solution 31P-NMR. . .:" as they change of topics from that point, shifting from the description of how the samples were prepared to how the NMR spectra were obtained. Line 169 : Replace ". . .test to test foridentify. . ." by " . . .test to identify. . . "

A: We changed them as suggested.

Q: Lines 183-184 : What more direct evidence have the authors that the third peaks in the fractograms could correspond to particle previously attached to the mebrane during focus time ?

A: The third peaks occurred after cross flow was zero. There was no force to bring the particles closed to the membrane. Therefore, the former particles attached to the membrane have possibly been eluted from the channel by the carrier flow.

Lines 195-196 : Is the claim made here that the nano-sized colloids from the cambisol contain P, Fe and Al in lower (negligible) concentrations compared to the same fraction in the stagnosol so true ? Indeed, I calculated the OC/Al and OC/Fe ratios of both soil types and they are not so different: 58 and 93 for Al, and 74 and 105 for Fe. Everything looks like if the nano-sized colloid fractions were equivalent in composition in both soil types, the fraction being simply more concentrated in the stagnosol compared to the cambisol.

A: It is correct. We also just mention that there were higher concentrations of OC, P, Fe, and Al in the nano-sized colloids from the Stagnosol compared to the Cambisol. The

concentrations here were the ratio of elemental mass to bulk soil mass. We indicated it in the Tables.

Line 202: Replace "Stagnols" by "Stagnosol"

A: We changed it as suggested.

Line 205: I agree that the UV signal is consistent with the OC peak distribution. However, I once again wonder about the reason why the UV peaks are shifted to somewhat lower elution time compared to the OC ones. Could the authors comment on that and provide explanation for this shift?

A: That is because the AF4 is connected with UV detector and the OCD detector were then connected with UV. The slight delay among the two peaks is due to the different length of tubes to UV and OC detectors which cause slightly different internal volume and retention time (see lines 587-589).

Q: Line 208-209: I agree that the second-size fraction of the stagnic Cambisol present the highest Fe, A, Si and P concentrations of the three analyzed second size-fractions. Considering however ratios of OC/Al, OC/Fe, and OC/P

A: The concentrations here mean different elements amount per kg soil not per kg OC. We indicated it in the Tables.

Q: Line 218 : It is not clear to me why OC sorbed on iron oxides materials should be of nanometric size? Could the authors cite papers which prove this to be so.

A: We found more OC in the first peak fraction, and assumed this additional OC was partly derived from the OC formerly sorbed on iron oxides. When iron oxides were dissolved, this sorbed OC was released as nano-size particles and was eluted in the first peak. We do not say all the OC sorbed on iron oxides should be of nanometric size. We think a part of nano-size OC in the Stagnosol was derived from the OC formerly sorbed on iron oxides. As found in our previous study with soil colloids of Luvisols (Jiang et al., 2015), higher OC content were found in the first peak fraction after the

dissolve of Fe oxides with DCB treatment. Lines 247-249: Correspondingly, the disso-lution of Fe oxides in the second fraction under stagnant water may also liberate OC from the organo-Fe mineral associations, thus releasing some OC to the nano-sized first fraction (Jiang et al., 2015a).

Q: Lines 237-238 : "This implies that the assignment of stagnic properties is related to its behavior in the colloidal particles and dissolved fraction". I find this sentence badly constructed. Do the authors mean that soils are classified according to the composition of the colloids they can release? I cannot believe that.

A: We changed it as follows: Lines 272-273: This implied that the stagnic properties have a greater impact on the colloidal particles and "dissolved" fraction compared to bulk soil.

Q: Line 243 (as regards Table 2): I wonder why TOC concentrations were not measured in the <300 KDa and < 5KDa fractions. Could the authors give an explanation for that?

A: The OC detector cannot give valid values to distinguish <300 kDa and <3 kDa frac-tions because of the extremely low concentration of OC in 3-300 kDa fractions.

Q: Lines 280-281. I do not see how Fig. 1 can be used to infer the proportion of clay-Fe oxides-OC-P associations in the 300 KDa-450 nm fractions. Could you explain?

A: The FFF results showed that there were Si, Al, Fe, OC and P in the 300 kDa-450 nm fraction. We measured another arable soil sample with TEM in former work (Jiang et al., 2015) and found clay minerals and Fe oxides in these fractions from soil (see lines 145-146).

Q: Line 300: The statement made here that the majority of P in the <3KDa fraction of the Cambisol was Po is quite "funny" in the light of what is said page 11 about the fact that the absence of NaOH-Na2EDTA most of the Pi is removed from the solution through sorption on the soil mienrals. Quite clearly, the data cannot be used to as-sess the proportions of Pi and Po as the methodology used biased these proportion.

They just can be used to inventory the organic species present, which is already an innovative and very important objective.

A: We agree and changed this sentence to: the majority of observed P in the < 3 kDa fraction of the Cambisol was organic P (lines 340-341).

Q: Line 348-352. What direct poof do the authors have that pyrophosphates is bound to Fe oxides? Could and alternative interpretation be that orthophosphates form ternary complexes with Fe3+ or Fe2+ ions themselves bound to OC?

A: Here Fe/Al means ions and we will emphasis it in the main text as follows: Lines 388-393: Considering that a high proportion of pyrophosphate (38.5%) existed in the 3-300 kDa fraction of the Stagnosol, which contained P mainly in OC-Fe(Al)2/3+-P associations (see above), it seems reasonable to assume that pyrophosphate existed as a colloidal OC-Fe(Al) 2/3+-pyrophosphate complex. In this regard, the accumulation of pyrophosphate may have been favored by the larger OC contents in this soil (Fig. 2 C).